# Orbitofrontal neurons acquire responses to 'valueless' Pavlovian cues during unblocking

Michael A McDannald[1]*, Guillem R Esber[1], Meredyth A Wegener[2],
Heather M Wied[3], Tzu-Lan Liu[4], Thomas A Stalnaker[1], Joshua L Jones[5],
Jason Trageser[6], Geoffrey Schoenbaum[1,5,7]*

[1]Intramural Research Program, National Institute on Drug Abuse, Baltimore, United States; [2]Center for Neuroscience Graduate Program, University of Pittsburg, Pittsburg, United States; [3]Program in Neuroscience, University of Maryland School of Medicine, Baltimore, United States; [4]Department of Psychology, National Taiwan University, Taipei, Taiwan; [5]Department of Anatomy and Neurobiology, University of Maryland School of Medicine, Baltimore, United States; [6]Department of Psychological and Brain Sciences, Johns Hopkins University, Baltimore, United States; [7]Department of Neuroscience, Johns Hopkins University, Baltimore, United States

**Abstract** The orbitofrontal cortex (OFC) has been described as signaling outcome expectancies or value. Evidence for the latter comes from the studies showing that neural signals in the OFC correlate with value across features. Yet features can co-vary with value, and individual units may participate in multiple ensembles coding different features. Here we used unblocking to test whether OFC neurons would respond to a predictive cue signaling a 'valueless' change in outcome flavor. Neurons were recorded as the rats learned about cues that signaled either an increase in reward number or a valueless change in flavor. We found that OFC neurons acquired responses to both predictive cues. This activity exceeded that exhibited to a 'blocked' cue and was correlated with activity to the actual outcome. These results show that OFC neurons fire to cues with no value independent of what can be inferred through features of the predicted outcome.

*For correspondence:
mcdannal@bc.edu (MAM);
geoffrey.schoenbaum@nih.gov (GS)

Competing interests: The authors declare that no competing interests exist.

## Introduction

The orbitofrontal cortex (OFC) is often described as signaling either an outcome expectancy, implying a knowledge of the features of the impending outcome (*Schoenbaum et al., 1998*; *Delamater, 2007*; *Ostlund and Balleine, 2007*; *Steiner and Redish, 2012*; *Luk and Wallis, 2013*), or a value that exists independent of those features (*Padoa-Schioppa, 2011*; *Levy and Glimcher, 2012*). Support for such pure or abstract value encoding comes largely from reports that single unit activity and the blood-oxygen level dependent (BOLD) response in the OFC tracks value, independent of outcome features such as identity or location or even the response required to obtain the outcome (*Padoa-Schioppa and Assad, 2006*; *Plassmann et al., 2007*; *Levy and Glimcher, 2011*).

Yet in some cases, outcome features could still be the underlying basis of apparent abstract value signals. This is conceivable even if the signal correlates with value across different outcomes, since some features or feature combinations might vary with value across the limited number of outcomes used in any particular session (but see *Padoa-Schioppa and Assad, 2006*, supplemental, and our correction notice). Further, any neural element (voxel or single unit) may participate in ensembles responding to more than one feature, so it is also possible that a particular element that appears not to distinguish specific features of different outcomes is in fact coding independent features that co-vary with each outcome's value.

**eLife digest** Imagine you are at a restaurant and the waiter offers you a choice of cheesecake or fruit salad for dessert. When making your choice it is likely that you will consider the features of these desserts, such as their taste, their sweetness or how healthy they are. However, when you decide which dessert to have, you will pick the one that you judge to have the highest value for you at that moment in time. In this sense, 'value' is a subjective concept that varies from person to person, while 'features' remain relatively static.

It is generally agreed that the orbitofrontal cortex (OFC) is involved in making these sorts of decisions, but its role is still a topic of debate. According to one theory the neurons in the OFC signal the subjective value of an outcome, whereas a rival theory suggests that they signal the features of the expected outcome. However, it has proved challenging to test these theories in experiments because it is difficult to say for certain that a given decision was clearly due to the value or a feature.

Now, McDannald et al. have devised an approach that can tell the difference between neurons signaling value and neurons signaling features. They trained thirsty rats to associate different odours with either an increase in the amount of milk they were given (a change in both value and a feature), or a change in the flavor of the milk (a change in a feature without a change in value). Extensive testing showed that the rats did not value one flavor over the other.

McDannald et al. then examined how the neurons in the OFC responded. If these neurons signal only value, they should only fire when the value of the outcome changes. On the other hand, if they signal features, they should fire when a feature changes, even if the value does not. It turned out that the neurons in the OFC responded whenever the features changed, irrespective of whether or not the value changed. These findings present a challenge to popular conceptions of the role of the neurons in the OFC.

So how can we address whether the OFC signals features of impending outcomes vs value independent of those features? One way is to strip away or 'block' the value portion of the outcome during learning, while leaving unblocked—free to enter into associations—the outcome's unique sensory and other features. This can be done by pairing a 'target' cue with a rewarding outcome in the presence of a cue that has been previously trained to predict a differently-flavored, but similarly-valued outcome. When this is done, the previously conditioned cue predicts the general value that is common to the two outcomes, but does not predict the unique features that distinguish the new outcome (note features are not limited to sensory properties, but might include the outcome timing, location, temperature, size, number, etc). As a result, the target cue acquires associations with the unique features of the new outcome but not with its general or common currency value (*Rescorla, 1999*; *Burke et al., 2008*). If OFC neurons represent only a general or common currency value, divorced from features, then they should respond no more to such a target cue than to a completely blocked cue (*Kamin, 1969*). However, if OFC neurons represent outcome features, independent of value, then they should respond to the target cue just as they do to a cue that has been explicitly unblocked by increasing the amount of the outcome delivered. Indeed, both pure value and outcome expectancy accounts of OFC function would predict neural activity to such an unblocked cue, but only an outcome expectancy account predicts encoding of the target cue signaling a valueless change in outcome flavor.

## Results

We recorded single–unit activity in the OFC in six rats during an odor-based unblocking task (*Figure 1A*). Prior to implantation with electrodes rats were trained to sample an odor in a central port following house light illumination and then respond to a reward well below for two drops of Nestlé's flavored milk (chocolate or vanilla, counterbalanced). This training was meant to establish the initial odor as a reliable predictor of a specific flavor and number of drops of milk. Each rat had extensive experience with both flavors, thus neither flavor was novel. Following initial training rats were implanted with microelectrodes in the OFC. When recovered from surgery, rats were retrained on the initial odor; after retraining, each rat underwent 7–9 rounds of unblocking.

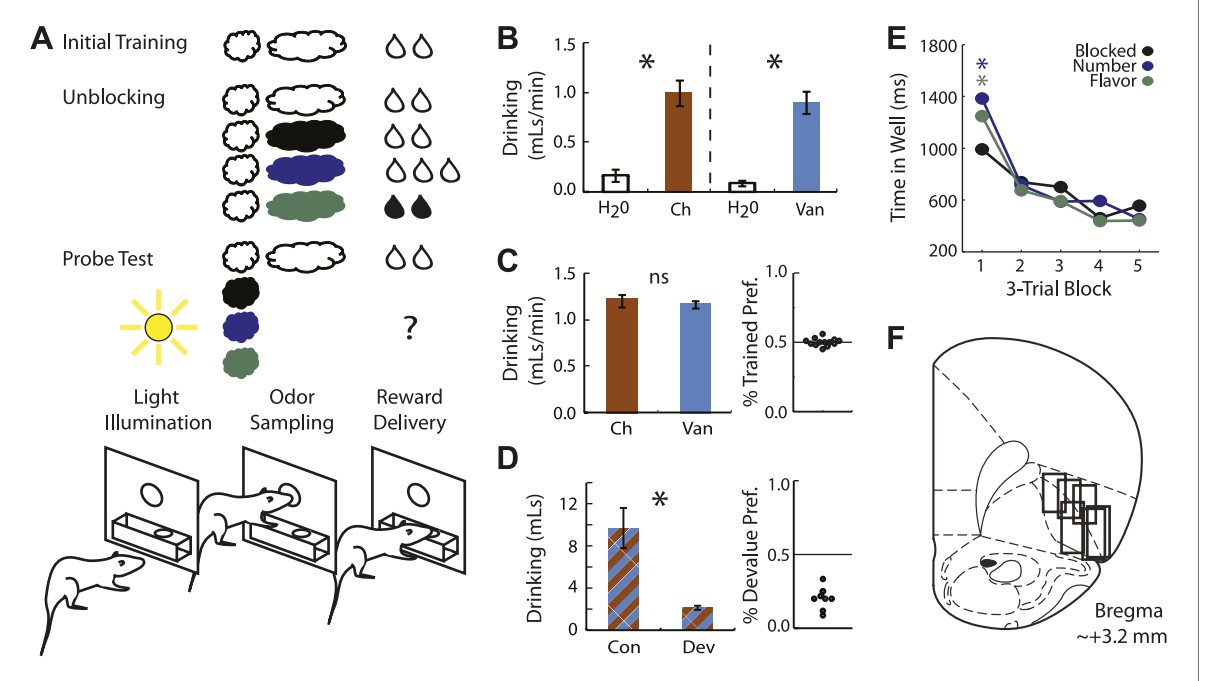

**Figure 1**. Experimental outline, behavior summary and recording sites. (**A**) Thirsty rats were initially trained to enter an odor port following illumination of a house light and respond at reward well below for two drops of flavored milk. Unblocking sessions consisted of four trial types. The first was a reminder of initial training. On the remaining three trial types, the originally trained odor was briefly presented followed by 1 of 3 novel odors. The reward following the novel odors was either unchanged (black; blocked trials), increased in number (blue; number trials), or its flavor was altered (green; flavor trials). Learning was assessed in a probe test in which the novel odors were presented in isolation, without reward. (**B**) 10-min consumption testing between chocolate and water, and vanilla and water on non-training days found that both were significantly and equally preferred to water (ANOVA, $F_{1,5} > 5$, p's < 0.05). (**C**) 2-min preference testing between chocolate and vanilla immediately following unblocking sessions found no flavor preference (t test, p > 0.1). Scatter plot (right) shows preference for the trained flavor on each individual test (n = 13). (**D**) 20-min consumption testing from a separate group of rats (n = 8) that received selective devaluation of one of the flavors found a significant difference in consumption between the non-devalued (Con) and devalued (Dev) flavors (t test, p < 0.01). This was true for every rat tested (right). (**E**) Time in the reward well is plotted for the probe test trials. ANOVA for time spent in the reward well with odor (blocked, number and flavor) and trial (1–15) as factors found a significant odor x trial interaction ($F_{1,47} = 3.45$, p < 0.05). Planned comparisons confirmed that on the first three trials rats spent significantly more time in the reward well following number and flavor odors compared to blocked (p's < 0.05) but responding to number and flavor did not differ (p > 0.1). (**F**) Single unit activity was recorded from the lateral orbital and agranular insular cortices at roughly 3.2 mm anterior to bregma. *p < 0.05; ns = not significant.

The following figure supplement is available for figure 1:

**Figure supplement 1**. Selective conditioned flavor aversion preference data.

Each round of unblocking began with 2 days of training and consisted of four trial types. One type was a reminder; the initially trained odor was followed by the expected outcome. On the other three trial types (blocked, number, flavor), rats were presented with the initially trained odor, followed immediately by one of three novel odors. On 'blocked' trials, the novel odor was followed by the expected two drops of the same flavor used in initial training. This outcome is fully predicted by the initial odor, thus the novel odor should be blocked from acquiring associative significance (**Kamin, 1969**). On 'flavor' trials, the novel odor was followed by two drops of the flavor not used in initial training (i.e., chocolate or vanilla). Here the value is unchanged, since the two flavors are equally preferred and the same amount is delivered, but the features of the outcome are different. Thus the novel target odor should enter into associations with the unique features of the outcome (**Rescorla, 1999**; **Burke et al., 2008**). On 'number' trials, the novel odor was followed by an additional drop of the flavor used in the initial training. Since the initial odor does not predict anything after the second drop, the novel odor should enter into associations with both the features and the additional value of the outcome presented in the third drop (**Holland, 1984**).

To ensure that learning in the flavor condition did not result from an explicit shift in value, two types of preference tests were administered in conjunction with the unblocking training procedures. In one test, given on days separate from unblocking, preference for each flavor over water was assessed. Both flavors were highly preferred to water (*Figure 1B*), demonstrating both are highly palatable. The more important preference test came just following unblocking sessions, from which the critical neural data came. In these tests, the two milk flavors were pitted directly against one another. This test is critical to demonstrate that specific satiety to one flavor did not develop over the course of the unblocking session, a finding that would strongly suggest different valuation of the two flavors. Indeed, we found no evidence of a preference in these tests (*Figure 1C*). In both types of tests the locations of the bottles were swapped every 20–30 s, meaning that rats were required to switch locations if the solution did in fact differ in value. This pattern was present when either milk flavor was compared to water but was absent when the two flavors were directly compared. The consumption data demonstrate that both flavors were highly palatable yet of equivalent value. Finally, to ensure the two flavors were discriminable we subjected another set of rats to a selective conditioned flavor aversion procedure. After initial exposure to both milk flavors, consumption of one flavor (fully counterbalanced) was devalued by pairing with LiCl-induced nausea while consumption of the other was paired with saline injection that is of minimal consequence. At no point did rats show a preference for the chocolate or vanilla flavor but all rats selectively reduced consumption of the devalued flavor. This was apparent both in conditioning (*Figure 1—figure supplement 1*) and in the final choice test (*Figure 1D*). Thus extensive consumption testing indicated that the flavors used were of equivalent value but readily discriminable.

In the unblocking sessions rats were sensitive to presentation of the novel odors, exhibiting longer latencies to respond at the reward well following odor sampling on these three trial types. Longer latencies to the novel odors were most apparent on the very first trial of each session, particularly on day 1. In support, ANOVA revealed a main effect of trial ($F_{1,47} > 2$, p's < 0.01) and a trial × day interaction ($F_{1,47} = 34.73$, $p < 0.01$). However the rats also learned that the two novel odors that predicted changes in the outcome were meaningful. This was evident in the extinction probe test in which they initially spent more time in the fluid well following sampling of the flavor and number odors than following the blocked odor (*Figure 1E*).

We recorded 240 single units during the first day of unblocking and 220 units on the second unblocking day in 48 rounds of training across all six rats (*Figure 1F*). To address our hypothesis, units from both unblocking days were screened for phasic responses to one of the four odors using a *t* test, which compared firing rates during the ITI and novel odor period (significance level = p < 0.0125; Bonferroni correction). This screen found 135 units (Day 1 = 79, Day 2 = 56) that showed a significant increase in firing to at least one of the odor cues. The majority–98/135 or 73% of the neurons within this population–exhibited activity that fell into one of two categories (see *Figure 2—figure supplement 1* for analysis of other neurons). We will consider each category in turn below.

The first major category, not directly anticipated by our hypothesis, consisted of neurons (55/135, *Figure 2A*) that showed a significant phasic response to each of the four odor cues. Since these neurons fired to all of the odors, even the blocked odor, their firing cannot be easily explained as signaling information about the predicted outcomes. However they might be signaling information about the cues themselves, such as their shared sensory features or intrinsic salience. Unlike shared sensory features, salience should be higher for the novel cues than for the pre-trained, initial odor in our design, and this pattern should be most noticeable early in training when the novel cues were first presented. As a population, these neurons did show greater activity to the blocked, number and flavor odors than to the more familiar, initial odor (*Figure 2B*). This effect was present despite the fact that we did not select based on this criterion. Further analyses of the individual units showed that many exhibited significantly higher firing to the novel odors than to the initial odor (*Figure 2C*), and few exhibited differences in firing among the three novel odors (*Figure 2D,E*). Moreover, when we examined the firing of neurons in this population on the first 10 trials of unblocking (31/79 odor-responsive neurons; *Figure 2—figure supplement 2*), analyzing the difference in firing between the novel and initial odor cues in a sliding, 300-ms window across each trial, we found that activity in this population was maximal at the onset of the novel odors and on the first exposure and then declined rapidly on subsequent trials (*Figure 2F*). This same pattern held when each novel odor was analyzed separately (*Figure 2—figure supplement 3*). Further, this pattern was only seen on the first day of

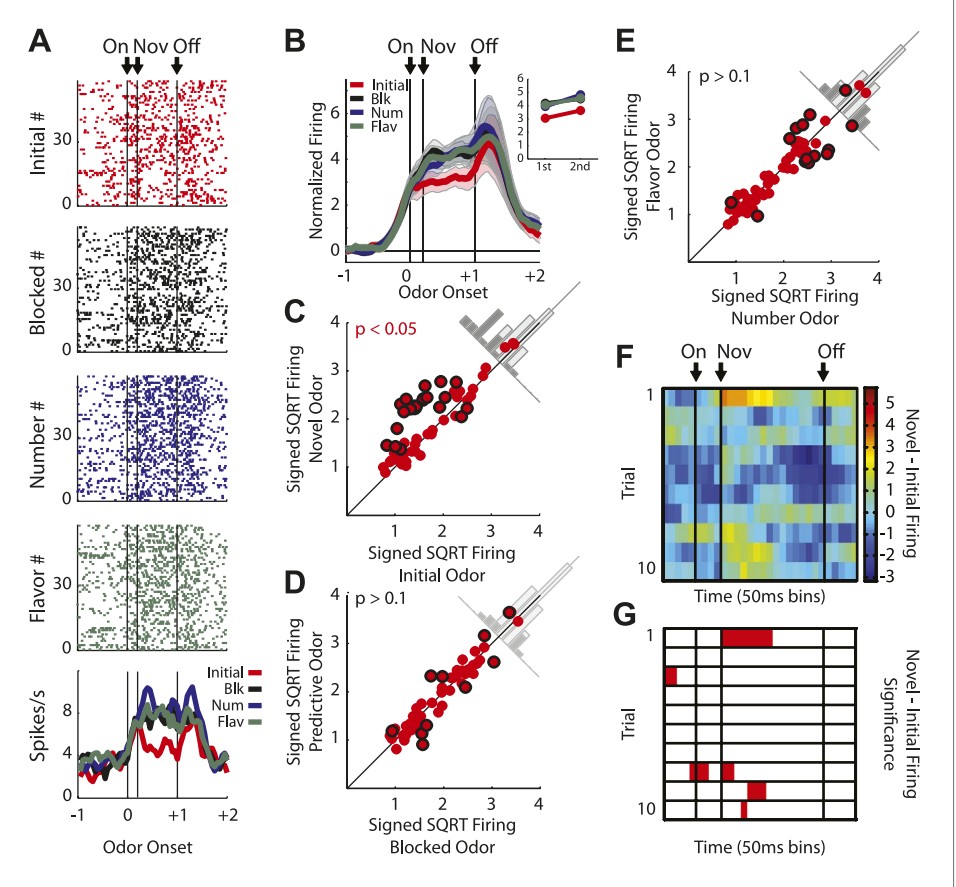

**Figure 2**. Single unit and population firing of putative salience neurons. (**A**) Raster plots for firing of a single unit are shown for all initial (red), blocked (black), number (blue) and flavor trials (green). Odor on (On) is indicated by the first vertical line, onset of novel odor (Nov) by the second vertical line and odor offset (Off) by the third. Each tick represents a spike. Average activity across all trials for each odor is plotted (bottom). (**B**) Mean neural activity (novel odor epoch–ITI) for the putative salience neurons (n = 55) is plotted. Line color as indicated in raster plots; shaded areas indicate standard error of mean. ANOVA with bin and odor as factors found significant effects of bin, odor and the bin × odor interaction ($F_{1,54} > 2.0$, p's < 0.01). ANOVA restricted to the novel odor period with odor and time (first 500 ms vs second 500 ms, shown in upper right inset) as factors found only a main effect of odor ($F_{1,54} = 13.0$, p < 0.01). Significant firing to the novel odors over the initial odor was observed throughout the novel odor period. (**C**) A scatter plot of novel odor firing vs initial odor firing is shown for putative salience neurons (n = 55). A signed square root transformation of firing was used to best visualize population spread; all statistics were performed on non-transformed firing rates. Individual neurons showing significant differences in firing between the odors are outlined in black (t test, p < 0.05). A non-parametric sign test found significant, preferential firing to the novel odors (Z = 3.24, p < 0.01). The population bias towards novel odor firing is apparent in the bar histogram aligned to the diagonal axis; on which the difference score for each neuron is plotted. Light gray bars represent units showing no differential firing; dark gray bars represent units showing significant differential firing. (**D**) A scatter plot of predictive vs blocked odor firing is shown. A sign test found no differential firing to the predictive and blocked odors by the putative salience population (Z = 0.54, p > 0.1). (**E**) A scatter plot of flavor and number odor firing is shown. A sign test found no differential firing to the number and flavor odors (Z = 0.27, p > 0.1). (**F**) Differential firing to the novel odors vs the initial odor on the first 10 trials of the first unblocking day was calculated and plotted for the putative salience population (n = 31). Differential firing was calculated in a 300-ms sliding window for each 50-ms bin moving away from novel odor onset: (mean [(blocked odor–ITI) + (number odor–ITI) + (flavor odor–ITI)]–initial odor–ITI). The difference score for each bin was then plotted, with dark red bins indicating maximal differential firing to the novel odors and dark blue indicating the opposite pattern (y-axis shown on right of heat plot). (**G**) The significance of the increased firing to the novel odors was determined by performing a one-tailed t test, comparing increases in differential firing to 0, using a significance of p < 0.05 and a sliding window as in (**F**). Red bins indicate significant elevations in firing to the novel odors over the initial odor.

*Figure 2. Continued on next page*

*Figure 2. Continued*

The following figure supplements are available for figure 2:

**Figure supplement 1**. Odor-responsive units not included in primary analyses.

**Figure supplement 2**. Firing of putative salience neurons on unblocking day 1.

**Figure supplement 3**. Heat plots for putative salience neurons for each novel odor.

**Figure supplement 4**. Heat plot for putative salience neurons on unblocking day 2.

unblocking (*Figure 2—figure supplement 4*). This pattern of activity is consistent with signaling of the salience of these cues.

The second major category, of greater relevance to our hypothesis, consisted of neurons (43/135, *Figure 3A–C*) that showed a significant phasic response the flavor and/or number odors (but did not fire to all four odors). Activity across this population was greater in response to the two predictive odor cues than to either the blocked or initial odors (*Figure 3D*), and an analysis of individual units showed that nearly all of these neurons (38/43) fired more to the predictive odors than to the blocked one (*Figure 3E*). This result marks these neurons as candidates for encoding of associative information or meaning, since this is the primary feature that distinguishes the two odor cues from the blocked odor.

Interestingly, these neurons did not appear to distinguish, at least as a population, between the flavor and number odors. For example, they showed similar levels of activity in response to both the flavor and the number odor (*Figure 3D,F*), and when we examined the firing of neurons in this population on the first 10 trials of unblocking (25/79 odor-responsive neurons, *Figure 3—figure supplement 1*), we found that differential firing to each cue developed at a similar rate during training (*Figure 3G–J*). The acquisition of differential firing to the flavor and number odors demonstrates that selective odor encoding was not driven by physical properties of the odors. If neurons were encoding the odor itself, independent of its outcome signaling, this would have been apparent on the very first trial. Thus, as a population, these neurons responded more strongly to the unblocked 'flavor' and 'number' odors than to the 'blocked' odor. Even more striking, the population responded similarly to a cue signaling a valueless change in the outcome flavor as they did to a cue that signaling that more of the outcome would be delivered.

Similar numbers of neurons fired to the flavor and number cues (flavor: 31, number: 37; $\chi^2 = 0.3$, $p = 0.47$). In our design, firing to the flavor cue cannot be readily explained as signaling general or common value. Thus these data confirm that many OFC neurons signal associative meaning independent of at least a general value. In support of this, odor firing in the flavor population, as well as the number population, was positively correlated with firing to the actual outcome delivered on each of these trials (*Figure 4*). This relationship supports the idea that cue-evoked activity in the flavor population is signaling features of the new outcome.

## Discussion

Neural signals in the OFC are often described as representing either outcome expectancies or abstract value. Although many studies have argued for one or the other, few have used behavioral designs that clearly dissociate predictions of these two hypotheses. Here we tried to address this question by using an unblocking procedure to strip away or 'block' the abstract value of the outcome during learning, while leaving unblocked—free to enter into associations—the outcome's sensory and other unique features. This approach revealed two distinct populations of OFC neurons.

One population consisted of neurons that fired immediately on initial presentation of all three target cues, perhaps reflecting these cues' novelty or salience. While unexpected and not directly relevant to the question motivating this study, this finding is consistent with reports of neural correlates of salience in OFC (*Kahnt and Tobler, 2013*; *Ogawa et al., 2013*) and with studies implicating the OFC in phenomena such as latent inhibition, set formation, and even auto-shaping (*Chudasama et al., 2003*; *Schiller and Weiner, 2004*; *Chase et al., 2012*) which depend in part on the appropriate

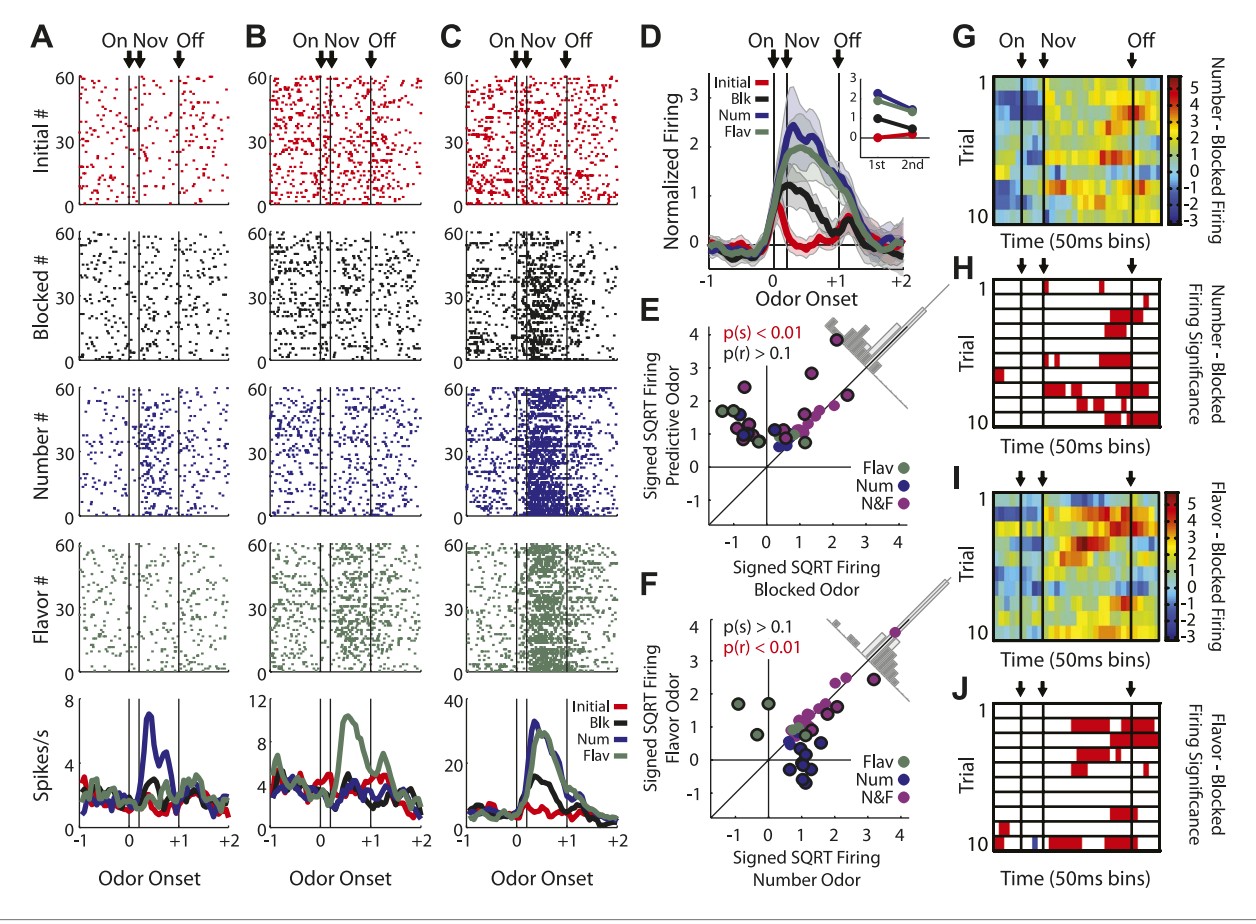

**Figure 3**. Single unit and population firing of putative predictive neurons. Single units plotted exactly as in **Figure 2A** showing (**A**) selective firing to the number odor (**B**) selective firing to the flavor odor (**C**) firing to both number and flavor odors. (**D**) Mean neural activity (novel odor epoch–ITI) for the putative predictive neurons (n = 43) is plotted. Meaning of line colors and shading is maintained. ANOVA with bin and odor as factors found significant effects of bin, odor and the bin × odor interaction ($F_{1,42} > 2.0$, p's < 0.01). ANOVA restricted to the novel odor period with odor and time (first 500 ms vs second 500 ms, shown in upper right inset) as factors found a main effect of odor ($F_{1,42} = 58.5$, p < 0.01) and an odor × time interaction ($F_{1,42} = 3.9$, p < 0.05). At both times firing to the predictive odors was significantly greater than the blocked and initial odors; blocked firing was greater than initial firing only in the first half. (**E**) A scatter plot comparing firing to the predictive odors (signed square-root transform) vs the blocked odor is shown for the predictive population (n = 43). Within this population there were three kinds of neurons based on firing vs ITI: number only (blue), flavor only (green) or flavor and number (purple). A sign test found significant, preferential firing to the predictive odors (Z = 4.88, p(s) < 0.01). Across all neurons there was zero correlation between predictive odor firing and blocked odor firing ($R^2 = -0.01$, p(r) = 0.39). (**F**) A scatter plot comparing firing to the flavor and number odors is shown for the predictive population (n = 43). While some neurons did show differential firing (outlined in black) to either the number (n = 14) or flavor (n = 4) odor a sign test found no bias in firing to the number or flavor odor across the entire population. (Z = 0.60, p > 0.1). Across all neurons there was a highly significant, positive relationship between number and flavor odor firing ($R^2 = 0.88$, p < 0.01). (**G**) Differential firing to the number odor vs the blocked odor on the first 10 trials of the first unblocking day was calculated and plotted for the number-responsive units within the predictive population (n = 21). This was done in as in (**Figure 2F**) except that the difference score was calculated as: (number odor–ITI)—(blocked odor–ITI). (**H**) Significance for increased firing to the number odor over the blocked odor was calculated as in (**G**). Red bins indicate significant elevations in firing to the number odor over the blocked odor. Blue bins would indicate significant decreases in firing to the number odor below the blocked odor. (**I**) Differential firing to the flavor odor vs blocked odor on the first 10 trials of the first unblocking day was calculated and plotted for flavor-responsive units within the predictive population (n = 18) as was done in (**G**). (**J**) Significance of differential firing calculated and displayed as in (**H**).

The following figure supplement is available for figure 3:

**Figure supplement 1**. Firing of putative predictive neurons on unblocking day 1.

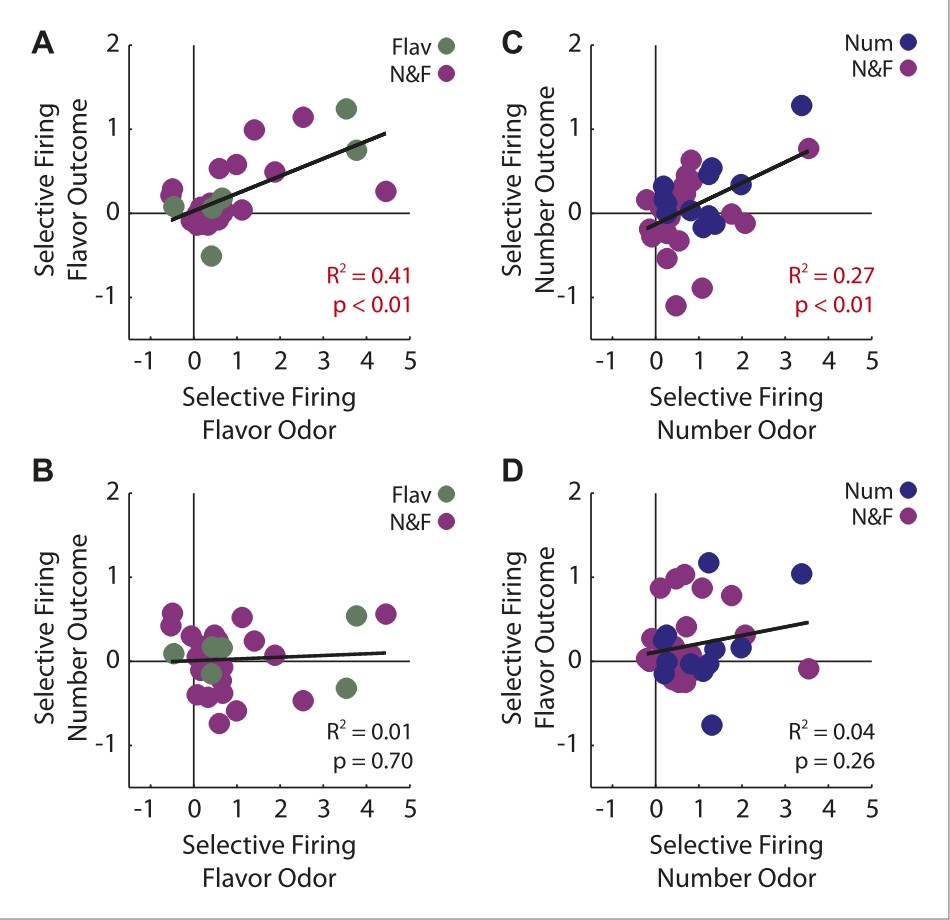

**Figure 4**. Outcome selectivity of predictive neurons. (**A**) For each predictive neuron that significantly increased firing to the flavor odor (total n = 31; flavor-only n = 6 [green]; number and flavor n = 24 [purple]) we plotted its selective firing to the flavor odor (x-axis; [normalized flavor odor firing – mean(normalized initial odor firing, normalized blocked odor firing, normalized number odor firing)]) against its selective firing to the flavor outcome (y-axis; [normalized flavor outcome firing – mean(normalized initial outcome firing, normalized blocked outcome firing, normalized number outcome firing)]). Comparison of odor firing of single neurons to the population found a single outlier (neuron firing was 3 stdev > population firing). The outlier was omitted from this analysis. There was a significant, positive relationship such that greater selective firing to the flavor odor was associated with greater selective firing the flavor outcome ($R^2$ = 0.41, p < 0.01). (**B**) This relationship was restricted to the flavor outcome; plotting selective flavor odor firing against selective number outcome firing revealed zero correlation ($R^2$ = 0.01, p = 0.70; calculation identical to A only mean(blocked, initial and flavor outcome firing) was subtracted from number outcome firing. (**C**) For each predictive neuron that significantly increased firing to the number odor (total n = 37; number only n = 12 [blue]; number and flavor n = 23 [purple]) we plotted it's selective firing to the number odor (x-axis; [normalized number odor firing – mean(normalized initial odor firing, normalized blocked odor firing, normalized flavor odor firing)]) against its selective firing to the number outcome (y-axis; [normalized number outcome firing – mean(normalized initial outcome firing, normalized blocked outcome firing, normalized flavor outcome firing)]). Two neurons showed selective odor firing 3 stdev above the population mean and were excluded from analysis. There was a significant, positive relationship such that greater selective firing to the number odor was associated with greater selective firing the number outcome ($R^2$ = 0.27, p < 0.01). (**D**) This relationship was restricted to the number outcome; plotting selective number odor firing against selective flavor outcome firing revealed zero correlation ($R^2$ = 0.04, p = 0.26) calculation identical to **C** only mean(blocked, initial and number outcome firing) was subtracted from flavor outcome firing. Finally, these statistical patterns were maintained if the flavor-only and number-only neurons were analyzed in isolation: (**A**) $R^2$ = 0.71, p = 0.03, (**B**) $R^2$ = 0.01, p = 0.82, (**C**) $R^2$ = 0.48, p = 0.01 and (**D**) $R^2$ = 0.18, p = 0.17. Flav = flavor, Num = number, N&F = number and flavor.

attribution of salience to cues. Together these results point to a largely unappreciated role for this area in the modulation of attention for the purposes of learning (*Esber et al., 2012*). Of course novelty is just one instance of 'salience'. Modern learning theories describe salience as a function of both intrinsic and acquired properties. Within this framework, this population was correlated with intrinsic salience.

The second population, of more direct relevance to our hypothesis, consisted of neurons that fired preferentially to the target cues that predicted changes in the outcome flavor. Nearly all of these neurons fired more to these cues than to the similarly trained but fully blocked cue, and this activity was acquired with learning, a pattern consistent with signaling of the associative significance or meaning of these cues (or possibly their acquired salience). Importantly, the acquired neural activity was observed to both the explicitly unblocked 'number' cue as well as to the 'flavor' cue, which was unblocked by shifting the features of the expected outcome while holding the value constant. This was accomplished by using two differently-flavored but similarly-preferred outcomes (*Figure 1C,D*). The lack of any flavor preference makes it unlikely that the cue added prior to this manipulation acquired what might be termed a general or cached value. Indeed, in prior work we have found that responding to this target cue in the probe test is dissociable from even the smallest animal-by-animal differences in conditioned or unconditioned responding to the two outcomes used (*Schoenbaum et al., 2011*), indicating that it is not driven by any sort of shift in value that might accrue to the cue.

Instead conditioned responding to a target cue unblocked by shifting the identity of the outcome seems to be particularly dependent upon the unexpected outcome's unique features, at least in comparison to an explicitly unblocked cue. As evidence of this, it has been shown that cues trained in this manner support behavior that is more sensitive to (in fact completely dependent upon) the features of the predicted outcome—or value inferred through those features—than similar behaviors supported by cues directly paired with reward in isolation. For example, conditioned reinforcement supported by a normally trained cue is insensitive to devaluation of the predicted outcome (*Parkinson et al., 2005*); however if the cue is trained like the 'flavor' cue in the current experiment, then devaluation of the predicted outcome completely abolishes the ability of the cue to serve as a conditioned reinforcer (*Burke et al., 2008*). This result indicates that a cue trained in this manner has little or no intrinsic, cached or acquired value except what can be inferred through knowledge of the features of the outcome. This cue's special link to the sensory features of the outcome is also apparent in that such cues retain the ability to support Pavlovian-to-instrumental transfer when that transfer is specific to the outcome (*Rescorla, 1999*). Interestingly outcome-specific transfer is both insensitive to devaluation (*Holland, 2004*) and disrupted by OFC lesions (*Ostlund and Balleine, 2007*), results which are difficult to reconcile with the view that the OFC is directly involved in the representation of value.

That OFC neurons developed robust responses to such a valueless cue indicates that many OFC neurons—more than half of the population responsive to the acquired significance of the cues—represent associative features that must be, strictly speaking, independent of general or common value. Notably this population included neurons that fired only to the flavor cue as well as neurons that participated in both the flavor and number ensembles. Such dual encoding would be expected if, as suggested earlier, individual neurons are not labeled lines but participate in ensembles coding more than one outcome feature (flavor, number, temperature, location, timing, etc).

Of course some neurons fired preferentially to the valued, number cue. The firing of these neurons could reflect the general value that accrued to this cue during training. However such firing could equally well reflect associations with features of the additional outcome delivered on these trials, known to develop when additional rewards are delivered during unblocking (*Holland, 1984*). While it is impossible to say for sure, the similarities in the overall level of neural activity to the flavor and number cues, their similar rates of development with learning, and the finding that neurons do develop activity to a valueless cue make the latter explanation the most parsimonious.

If OFC neurons signal associations between cues and specific outcome features, this would accord well with results showing that the OFC is not necessary when behavior—or learning—can be accomplished using general value alone. For example, the OFC is not required for simple Pavlovian or instrumental conditioning (*Gallagher et al., 1999*; *Izquierdo et al., 2004*; *Ostlund and Balleine, 2007*; *Gremel and Costa, 2013*) discrimination learning (*Schoenbaum et al., 2002*; *Izquierdo et al., 2004*; *McDannald et al., 2005*; *Walton et al., 2010*), extinction by reward omission (*Takahashi et al., 2009*),

transfer (*Ostlund and Balleine, 2007*), and even perhaps reversal learning (*Rudebeck et al., 2013*), all of which can be accomplished without reference to specific information about predicted outcomes. Similarly both blocking and unblocking—when it can be accounted for by value—do not require the OFC (*Burke et al., 2008*; *McDannald et al., 2011*). While the preserved function in these studies could reflect compensation by other areas, it must at least call into question the idea that OFC, writ large, is fundamental to all behavior that reflects value, instead highlighting suggestions that common value representation may at least be limited to the medial subregion (*Noonan et al., 2010*, *2012*). Indeed recent work in humans has shown that OFC represents specific outcome features and that more lateral orbital areas represent those outcomes in a way that is dependent upon prior cues (*Klein-Flugge et al., 2013*). Thus far from signaling general value about outcomes without regard to their features and attendant events, this work shows that the OFC maintains highly specific representations.

Such highly specific representations are consistent with observations that the OFC is necessary for superficially similar behaviors (Pavlovian or instrumental responding, discriminations, even learning) when they require knowledge of the outcome features in order to recognize errors or to derive or infer a value (*Gallagher et al., 1999*; *Izquierdo et al., 2004*; *Ostlund and Balleine, 2007*; *McDannald et al., 2011*; *Gremel and Costa, 2013*). This is even true in the current paradigm, where we have shown that the OFC is required for the development of conditioned responding to the target cue paired with a shift in outcome identity but not to the target cue paired with additional outcome (*McDannald et al., 2011*). Our present finding of robust encoding of a valueless Pavlovian cue that exceeds that of a blocked cue, and is equivalent to a cue paired with additional outcome, provides further support for outcome expectancy theories of OFC function.

## Materials and methods

### Subjects

Male Long-Evans rats were obtained at 200–250 g from Charles River Labs, (Wilmington, MA). Rats were tested at the University of Maryland School of Medicine and the NIDA-IRP in accordance with SOM and NIH guidelines (12-CNRB-108).

### Surgery and histology

Using aseptic, stereotaxic surgical techniques, a drivable bundle of 16, 25 µm diameter FeNiCr wires (Stablohm 675, California Fine Wire, Grover Beach, CA) was chronically implanted dorsal to OFC in the left hemisphere at 3.0 mm anterior to bregma, 3.2 mm laterally, and 4.0 mm ventral to the surface of the brain in each rat. Immediately prior to implantation, these wires were freshly cut with surgical scissors to extend ~1 mm beyond the cannula and electroplated with platinum (H2PtCl6; Aldrich, Milwaukee, WI) to an impedance of ~300 kΩ. At the end of the study, the final electrode position was marked by passing a 15 µA current through each electrode. The rats were then perfused, and their brains removed and processed for histology using standard techniques.

### Blocking task

Recording was conducted in grounded aluminum chambers approximately 18″ on each side with sloping walls narrowing to an area of 12″ × 12″ at the bottom. A central odor port was located above a fluid well on a panel in the right wall of each chamber. Two lights were located above the panel. The odor port was connected to an airflow dilution olfactometer to allow the rapid delivery of olfactory cues to the odor port; odors where chosen from compounds obtained from International Flavors and Fragrances (New York, NY). The fluid well was connected to lines controlling the independent delivery of the fluid rewards. Task control was implemented via computer running a behavioral program written in C++.

Prior to implantation with microelectrodes, rats were water deprived by restricting daily access to 1 hr following each training session. Water-deprived rats were progressively shaped to hold in the odor port for 1 s to receive two drops of water at the well. After shaping, rats received further training until they were proficiently responding for the initial odor in order to receive two boli of milk (vanilla or chocolate-flavored, counterbalanced); this involved as many as 15 sessions, with a maximum of 240 trials in each session. Proficient responding was characterized as correctly completing ~200 trials per session. Each trial began with house light illumination after which rats had 3 s to enter the odor port. Failure to enter the odor port resulted in restart of the trial. Once in the odor port rats were required to hold for 1 s and upon exit had 3 s to enter the reward well. Again, failure to hold for 1 s or enter the

reward well within 3 s resulted in restart of the trial. On alternate days, rats were given 20-min ad libitum exposure to the untrained milk flavor.

Following implantation, rats were retrained on the initial odor and once single units were isolated the unblocking procedure began. On the 2 learning days, rats received four trial types. The first was a reminder of initial training. The remaining trial types began with a 200 ms presentation of the initial odor and were followed by 1 of three 800 ms, novel yet distinguishable odors. The behavioral requirements of each of these trial types were exactly as in initial training. Rats completed between 30–60 trials involving each novel odor per session. On the subsequent probe test day, rats received a brief reminder of each trial type, ~10 total trials and then were presented the novel odors alone without reward, interleaved with trials in which the initial odor was presented with reward, in order to maintain responding. On the unrewarded, novel-odor extinction trials, the requirement to sample the odor for 1-s and respond to the reward well was lifted. The unblocking procedure was repeated seven to nine times per rat using a new set of blocked, number and flavor odors each time; in some cases, a new initial odor was also trained prior to repeating unblocking. When the initial odor was changed rats were trained on the new initial odor for 4–5 sessions prior to the first unblocking procedure with that odor.

## Consumption tests

Consumption tests were given in a housing cage separate from their home cage and experimental chamber. Two varieties of two-bottle tests were given. In the first, consumption of one of the flavored milks (chocolate or vanilla) was compared to consumption of water. These tests were 10-min in duration and occurred on days when unblocking training was not performed. The second test directly compared consumption of the two flavored milks. These tests were 2-min in duration and occurred immediately following unblocking sessions. This second test was critical for showing that rats did not become selectively sated to the flavored milk they experienced more in the unblocking sessions. Further, these tests occurred immediately after unblocking sessions while rats were still in the recording setting, informing any preference that would have developed over the course of the unblocking session. For all tests the location of the bottles was swapped roughly every 20–30 s to equate time on each side.

## Selective conditioned flavor aversions

Naïve rats were exposed to the vanilla and chocolate-flavored milk twice each in 1-hr sessions. During pre-exposure sessions intraperitoneal saline injections (0.9%, 5 ml/kg i.p.) were given to habituate rats to the injection procedure. Conditioning consisted of five, 1-hr exposures to each of the two solutions spaced over 10 days. Exposure to the devalued flavor was followed by injection of lithium chloride (0.3 M LiCl, 5 ml/kg i.p.). Exposure to the control flavor was followed by injection of saline. Consumption of each flavor was measured daily. A final 20-min choice test was given in which both flavors were present. At the 10-min mark the locations of the bottles were swapped. All factors (identity of devalued flavor, order of flavor presentation, side of flavor during choice test) were fully counterbalanced.

## Single-unit recording

Neural activity was recorded using two identical Plexon Multichannel Acquisition Processor systems (Dallas, TX), interfaced with odor discrimination training chambers described above. After recovery from surgery, electrodes were advanced daily until activity was obtained on a majority of wires. During this process, rats received reminder training using the pre-trained initial odor, as described above. Once the electrode was in a suitable location in OFC, single units were isolated and rats showed proficient responding, the rat began unblocking. During this 3-day procedure, the electrode was generally left in the same position. Thus, although we will not attempt to track neurons across sessions, the general population should be similar across each 3-day period. Following the completion of each 3-day unblocking procedure, the electrode was advanced 40–80 µm, and the process was repeated using new odor cues to test neurons in a new location in OFC.

## Statistical data analysis

Units were sorted using Offline Sorter software from Plexon Inc (Dallas, TX) using a template matching algorithm. Sorted files were then processed in Neuroexplorer to extract unit timestamps and relevant event markers. These data were subsequently analyzed in Matlab (Natick, MA). To examine activity to the novel odors, we examined activity from 300–1300 ms after initial odor onset, which corresponded

approximately to the time during which the novel odors were delivered to the odor port. To examine activity to the flavor outcome, we examined activity for 2000 ms starting with the first drop delivery. To examine activity to the number outcome, we examined activity for 2000 ms starting 1000 ms following first drop delivery, coinciding with the third drop delivery. The inter-trial interval was defined as the 2 s prior to illumination of the house light. Normalized firing was calculated by taking the firing rate during a period of interest minus firing rate during the ITI: Normalized firing = (Period spikes/s) − (ITI spikes/s). Neurons were identified as being odor responsive with a Bonferroni-corrected $t$ test (four separate tests, 0.05/4 = 0.0125; p < 0.0125) comparing elevations in firing during each of the four odor epochs (initial, blocked, number and flavor) from their respective ITIs. Neurons were classified as putative salience neurons if they significantly increased firing to all odors; putative predictive neurons were classified by significantly increasing firing to either or both the number and flavor odors, but not to all four odors. Single-unit and population activity was plotted in 50-ms bins; population activity was analyzed with repeated measures ANOVA with bin (50 ms) and odor trial (initial, blocked, number and flavor) as factors.

Heat plots were constructed by calculating difference scores between normalized firing to Novel and Initial odors (*Figure 2F*), Number and Blocked odors (*Figure 3G*) and Flavor and Blocked odors, in 200-ms sliding windows moving away from the novel odor onset in 50 ms increments. Warmer colors (dark red) indicated positive difference scores while cooler colors (dark blue) indicated negative difference scores. This was done for the first 10 trials of the identified population. Significance of differential firing to Novel and Initial odors (*Figure 2F*), Number and Blocked odors (*Figure 3G*) and Flavor and Blocked odors was determined by performing a one-tailed $t$ test comparing differential firing to zero in the exact same 200-ms sliding windows for each of the 10 trials. Finally, for purposes of visualization only; single unit and population firing, as well as heat plot comparisons were smoothed by taking a four-bin average moving in 50 ms increments. This applied to *Figure 2* (A–bottom, B, and F), *Figure 3* (A–bottom, B–bottom, C–bottom, D, G, I) and *Figure 4A*. All statistical analyses were performed on unsmoothed data.

## Acknowledgements

This work was supported by the Intramural Research Program at the National Institute on Drug Abuse and DA034010 (MAM). The opinions expressed in this article are the authors' own and do not reflect the view of the NIH/DHHS.

## Additional information

### Funding

| Funder | Grant reference number | Author |
| --- | --- | --- |
| National Institute on Drug Abuse | IRP Lab | Michael A McDannald, Guillem R Esber, Meredyth A Wegener, Heather M Wied, Tzu-Lan Liu, Thomas A Stalnaker, Joshua L Jones, Jason Trageser, Geoffrey Schoenbaum |

The funder had no role in study design, data collection and interpretation, or the decision to submit the work for publication.

### Author contributions

MAM, Conception and design, Acquisition of data, Analysis and interpretation of data, Drafting or revising the article; GRE, MAW, HMW, T-LL, TAS, JLJ, JT, Acquisition of data, Drafting or revising the article; GS, Conception and design, Analysis and interpretation of data, Drafting or revising the article

### Author ORCIDs

Meredyth A Wegener, http://orcid.org/0000-0003-0699-7748

### Ethics

Animal experimentation: Rats were tested at the University of Maryland School of Medicine and the NIDA-IRP in accordance with SOM and NIH guidelines (12-CNRB-108).

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
