## [Decision Letter]

Thank you for sending your work entitled “Orbitofrontal neurons acquire responses to 'valueless' pavlovian cues during unblocking” for consideration at *eLife*. Your article has been favorably evaluated by Eve Marder (Senior editor), a Reviewing editor, and 2 reviewers.

The Reviewing editor and the reviewers discussed their comments before we reached this decision, and the Reviewing editor has assembled the following comments to help you prepare a revised submission.

There has been a long, ongoing debate about whether the orbitofrontal cortex represents the value associated with cues or if these signals instead encode information relevant for learning and inference. Here, McDannald and colleagues test this directly by comparing cell responses in orbitofrontal cortex during two types of unblocking: one where the introduction of a novel cue after a familiar one is accompanied by an increase in the reward magnitude or a second type where a different novel cue is followed by a different flavored reward but of comparable value. This is contrasted with activity during blocking where a novel cue is followed by the same reward associated with the familiar cue. They study two major populations of OFC cells, one which responds to the novel cues on day 1, and a second which was selectively activated during both types of unblocking. This is an interesting manuscript that takes a novel approach to understanding frontal cortex function by creating cues with an associative significance but which do not indicate a value. The fact that neurons in orbitofrontal cortex (OFC) respond to these cues is interesting and suggests that the prevalent view that the OFC just codes value is incorrect. We therefore think that the manuscript will be of broad interest. In general it is well written and clear.

Major concerns:

1) Please provide degrees of freedom with the ANOVAs to know what exactly is being tested.

2) How consistent over sessions was the unblocking effect? If session is included as a factor (up to the minimum number that all rats run), does this interact with the odor x trial RT effect on probe trials? If there is any variance in the unblocking effect across animals and/or sessions, it would be interesting to know whether the magnitude of the unblocking effect related to the response of the neurons.

3) What is the precise psychological interpretation of increased RTs in the probe test? They look convincingly different (at least without error bars on the figure!) but it is not clear what this meant for what information the animal was taking from the unblocked or blocked cues.

4) It would be good to know what prompted the initial learned odor to be changed in some rats.

5) Some details of the spread of the neurons across the 6 animals would be useful to show that they are not all coming from a small sample of the animals.

6) An important finding is that there are similar numbers of cells responding to the change in reward identity and magnitude. However, the test seemed just to be on the overall numbers of significant cells and not about whether these are the same neurons coding for both types of outcome change or if there are instead separate, equally-sized populations that care about changes in reward magnitude and changes in reward identity.

---

## [Author Response]

*1) Please provide degrees of freedom with the ANOVAs to know what exactly is being tested*.

We apologize for omitting the degrees of freedom. This was an oversight on our part and they have now all been added for all ANOVAs.

*2) How consistent over sessions was the unblocking effect? If session is included as a factor (up to the minimum number that all rats run), does this interact with the odor x trial RT effect on probe trials? If there is any variance in the unblocking effect across animals and/or sessions, it would be interesting to know whether the magnitude of the unblocking effect related to the response of the neurons*.

When session is included as an ANOVA factor there is a main effect of session. That is, with continued probe tests the rats generally spend less time in the reward well during probe test. However, there is no interaction of session with odor. Responding is decreased to all odors across the board. With repeated extinction sessions rats seem to learn that in this phase of the task no reward will be given and extinguish much more rapidly with each successive probe test.

*3) What is the precise psychological interpretation of increased RTs in the probe test? They look convincingly different (at least without error bars on the figure!) but it is not clear what this meant for what information the animal was taking from the unblocked or blocked cues*.

We apologize for not making the results clearer. The dependent measure of probe test behavior was time spent in the reward well in the absence of reward. This is thought to reflect the degree to which rats expect a reward to be delivered.

*4) It would be good to know what prompted the initial learned odor to be changed in some rats*.

The initial odor was changed for purely technical reasons. The odor cartridges we use hold a maximum of 16 different odors. By the time rats had completed 4-5 unblocking sessions all odors from a single cartridge had been used. In order to complete more unblocking sessions, rats were trained on a new initial odor from a new cartridge and the novel odors from the cartridge used for the blocked, number and flavor odors.

*5) Some details of the spread of the neurons across the 6 animals would be useful to show that they are not all coming from a small sample of the animals*.

Here are the units per individual for the 6 rats composing the study, taken from both unblocking sessions 1 and 2:mm1052mm1543mm21124mm2280mm23121mm2440

Numerically the most units came from mm21; however, the primary results do not change if the units from mm21 are omitted from analysis.

*6) An important finding is that there are similar numbers of cells responding to the change in reward identity and magnitude. However, the test seemed just to be on the overall numbers of significant cells and not about whether these are the same neurons coding for both types of outcome change or if there are instead separate, equally-sized populations that care about changes in reward magnitude and changes in reward identity*.

This is an important aspect of the manuscript and one we attempted to address with the scatter plots in Figure 3. Figure 3 demonstrates that the vast majority of number and/or flavor responsive neurons show greater firing to that odor compared to the blocked odor – meaning this firing is associative in nature. Figure 3 is meant to precisely ask if this is because there are two separate populations of number and flavor neurons or if these come from a common population. While there were some number-responsive neurons (blue only neurons falling on the x-axis) and some flavor-responsive neurons (green only neurons falling on the y-axis), the majority of neurons were responsive to both odors (purple neurons). For the most part a single population coded for both the increases number and change in flavor.